# The Systemic Inflammation Response Index Efficiently Discriminates between the Failure Patterns of Patients with Isocitrate Dehydrogenase Wild-Type Glioblastoma Following Radiochemotherapy with FLAIR-Based Gross Tumor Volume Delineation

**DOI:** 10.3390/brainsci14090922

**Published:** 2024-09-15

**Authors:** Sukran Senyurek, Murat Serhat Aygun, Nulifer Kilic Durankus, Eyub Yasar Akdemir, Duygu Sezen, Erkan Topkan, Yasemin Bolukbasi, Ugur Selek

**Affiliations:** 1Department of Radiation Oncology, School of Medicine, Koc University, 03457 Istanbul, Turkey; ssenyurek@kuh.ku.edu.tr (S.S.); ndurankus@kuh.ku.edu.tr (N.K.D.); eakdemir@kuh.ku.edu.tr (E.Y.A.); dsezen@kuh.ku.edu.tr (D.S.); ybolukbasi@ku.edu.tr (Y.B.); 2Department of Radiology, Altunizade Acibadem Hospital, 03457 Istanbul, Turkey; drserhataygun@gmail.com; 3Department of Radiation Oncology, Faculty of Medicine, Baskent University, 01120 Adana, Turkey; docdretopkan@gmail.com

**Keywords:** glioblastoma, radiochemotherapy, failure pattern, systemic inflammation response index

## Abstract

Background/Objectives: The objective of this study was to assess the connection between the systemic inflammation response index (SIRI) values and failure patterns of patients with IDH wild-type glioblastoma (GB) who underwent radiotherapy (RT) with FLAIR-based gross tumor volume (GTV) delineation. Methods: Seventy-one patients who received RT at a dose of 60 Gy to the GTV and 50 Gy to the clinical target volume (CTV) and had documented recurrence were retrospectively analyzed. Each patient’s maximum distance of recurrence (MDR) from the GTV was documented in whichever plane it extended the farthest. The failure patterns were described as intra-GTV, in-CTV/out-GTV, distant, and intra-GTV and distant. For analytical purposes, the failure pattern was categorized into two groups, namely Group 1, intra-GTV or in-CTV/out-GTV, and Group 2, distant or intra-GTV and distant. The SIRI was calculated before surgery and corticosteroid administration. A receiver operating characteristic (ROC) curve analysis was used to determine the optimal SIRI cut-off that distinguishes between the different failure patterns. Results: Failure occurred as follows: intra-GTV in 40 (56.3%), in-CTV/out-GTV in 4 (5.6%), distant in 18 (25.4%), and intra-GTV + distant in 9 (12.7%) patients. The mean MDR was 13.5 mm, and recurrent lesions extended beyond 15 mm in only seven patients. Patients with an SIRI score ≥ 3 demonstrated a significantly higher incidence of Group 1 failure patterns than their counterparts with an SIRI score < 3 (74.3% vs. 50.0%; *p* = 0.035). Conclusions: The present results show that using the SIRI with a cut-off value of ≥3 significantly predicts failure patterns. Additionally, the margin for the GTV can be safely reduced to 15 mm when using FLAIR-based target delineation in patients with GB.

## 1. Introduction

The standard treatment for patients with newly diagnosed glioblastoma (GB) involves maximum safe resection followed by adjuvant radiation therapy (RT) along with concurrent and post-RT temozolomide (TMZ) [1,2,3]. Patients with GB commonly face poor prognoses, primarily attributable to an 80% incidence of local recurrence and an 8–30% likelihood of distant brain recurrences after the conclusion of standard therapy. Despite aggressive multimodal treatment modalities, the 2-year and 5-year survival rates remain at 17.2% and 5.5%, respectively [1].

Adjuvant radiotherapy (RT) is an indispensable component of standard treatment for GB. Nevertheless, notable differences in target delineation exist between the two widely recognized RT contouring guidelines, namely the guidelines published by the RTOG/NRG (Radiation Therapy Oncology Group) and the ESTRO (the European Society for Radiotherapy and Oncology) [4,5]. The RTOG/NRG’s methodology utilizes a two-phase approach. It commences by delineating gross tumor volume 1 (GTV1) based on T2-weighted images (T2w) or fluid-attenuated inversion recovery (FLAIR) alterations in MRI (Magnetic Resonance Imaging), followed by a 20 mm margin in all directions added to delineate clinical target volume 1 (CTV1). Subsequently, a dose of conventionally fractionated 46 Gy is prescribed to CTV1. Following this, GTV2 is identified by T1-weighted postcontrast (T1w), and a 20 mm margin is added to form CTV2, which receives an additional 14 Gy in the second phase. In contrast, the recently released ESTRO-EANO (European Association of Neuro-Oncology) guideline supports a single-phase treatment approach involving the delineation of a unified treatment volume comprising postoperative cavity, the enhancing T1w lesion, and T2w/FLAIR changes. This approach recommends utilizing 15 mm margins for the CTV and 3 mm margins for the planning target volume (PTV), with a prescribed total RT dose of 60 Gy. Diverging from these protocols, our institutional protocol has been utilizing FLAIR-based gross target delineation since 2009. This entails defining the GTV as the postoperative cavity, the enhancing T1w lesion, and any T2w/FLAIR invasion. Additionally, we employ a simultaneous integrated boost technique to administer 60 Gy to the GTV and 50 Gy to the CTV (CTV = GTV + 20 mm) throughout 30 fractions.

Identifying non-contrast-enhancing lesions on T2w/FLAIR is imperative because tumor cells can extend beyond the contrast-enhanced lesion into the surrounding brain tissue. Furthermore, certain high-intensity regions on T2w/FLAIR may possess malignancy comparable to that of contrast-enhanced lesions [6]. Evidence from the surgical series indicates that maximal resection of the non-contrast enhancing lesion yields an overall survival advantage, underscoring the necessity of incorporating this area into the GTV [7,8].

The development and progression of gliomas are intricately associated with intratumoral heterogeneity, genetic alterations, multiple molecular signaling pathways, inflammation, and immune response statuses [9,10,11,12], which make up the focus of the current study. Recently, systemic inflammation parameters have emerged as new prognostic markers in patients with GB, as GB represents the glioma with the highest inflammatory response and suppressed immunity [13,14]. These novel blood-based cellular parameters encompass the neutrophil-to-lymphocyte count ratio (NLR), systemic immune inflammation index (SII), and systemic inflammation response index (SIRI) [14,15,16,17]. As objective and replicable indicators, these parameters complement conventional markers, such as the Karnofsky Performance Status, tumor location, age at presentation, neurological status, Recursive Partitioning Analysis (RPA) group, and extent of surgery. Prior investigations have mainly focused on the prognostic implications of these biomarkers on the survival outcomes of patients with GBs and other gliomas [14,15,16,17]. However, research has yet to examine the correlation between systemic inflammation and patterns of treatment failure in these patients, particularly within the framework of GTV delineation employing FLAIR imaging. Therefore, our current study aimed to assess the patterns of failure in IDH wild-type GBs as per the aforementioned protocol and to examine the discriminative capacity of the SIRI across diverse brain relapse patterns.

## 2. Patients and Methods

### 2.1. Patients

For this retrospective study, we examined the medical records of patients diagnosed with histologically confirmed GB at Vehbi Koc Foundation Healthcare Institutions between March 2009 and May 2023. This study’s eligibility criteria encompassed patients with IDH wild-type GB who exhibited treatment failure and had undergone a sufficient number of follow-up MRI scans. Additionally, this study necessitated the presence of documented preoperative and pre-RT fusion MRIs, a complete blood count before surgery, and the status of corticosteroid usage. The exclusion criteria included patients who lacked corticosteroid usage data, those without an MRI performed at least 48 h before the RT initiation, patients without recurrence, cases of gliomatosis cerebri, brain stem tumors, and patients who underwent short-term RT. We additionally excluded patients who were immunocompromised, had renal or hepatic failure, had chronic immune suppressive diseases, were on immunosuppressant medications, refused concurrent or adjuvant TMZ, underwent hypofractionated RT schemes, or had incomplete treatment due to toxicity or active infection.

### 2.2. Target Delineation and Systemic Treatment

The initial stage of target delineation involved the co-registration of MRI and the planning of CT images, obtained with a 1 mm slice thickness. The patient’s head and neck were positioned in a neutral alignment for each imaging modality. The GTV encompassed the post-surgical resection cavity, any remaining enhanced tumor visible on contrast-enhanced T1w MRI, and hyperintense areas on T2w/FLAIR MRI. The CTV was established by adding a 20 mm margin to the GTV in all directions while respecting anatomical boundaries, such as the skull (utilizing the bone window), dura mater, falx, tentorium cerebelli, ventricles, visual pathways/optic chiasm, and brainstem, ensuring that the margin did not infringe upon the white matter tracts extending to these areas. The planning target volume (PTV) was created by extending a 2–3 mm margin in all directions from the CTV.

All patients received RT using a simultaneous integrated boost technique. The prescribed dose was 2 Gy per fraction for the GTV and 1.66 Gy per fraction for the CTV. This treatment was administered once daily, five days a week, over six weeks, resulting in a total dose of 60 Gy to the GTV and 50 Gy to the CTV. Concurrent TMZ was administered at a dose of 75 mg/m^2^/day during radiotherapy, and prophylactic trimethoprim-sulfamethoxazole against Pneumocystis jirovecii was prescribed during the entire course of concurrent treatment. Subsequently, adjuvant TMZ was administered at a dose of 150–200 mg/m^2^/day (5 days per cycle every 28 days) for 6–12 months.

### 2.3. Delineation of Failure Pattern

Patients underwent initial MRI scans six weeks post completion of RT, followed by subsequent serial scans at three-month intervals. Disease progression was defined according to the Response Assessment in Neuro-Oncology (RANO) criteria for high-grade gliomas [18]: a 25% increase in enhancing lesions, a significant increase in T2/FLAIR non-enhancing lesions while on stable or increasing doses of corticosteroids compared with the baseline scan, or the presence of any new lesion. Advanced imaging techniques, such as perfusion MR imaging, MR spectroscopy, and diffusion-weighted imaging (DWI), were utilized to improve the accuracy of diagnosing failures [19]. Perfusion MR imaging was mainly used to enhance lesions due to a high relative cerebral blood volume (rCBV), which is associated with tumor recurrence [20]. MR spectroscopy (MRS) was utilized to assess brain metabolites, specifically choline and N-acetyl aspartate (NAA), as an elevated choline/NAA ratio indicates disease progression [21]. Additionally, reduced apparent diffusion coefficients (ADCs) in diffusion-weighted imaging (DWI) were used to support the diagnosis [19]. In instances of recurrence, the MR images were fused with the planning CT to analyze failure patterns. An experienced neuroradiologist (M.S.A) assessed the maximum distance of recurrence (MDR) from the GTV in whichever extended farthest in the axial, sagittal, and chronal planes. The localization and extent of the recurrent tumor relative to the original treatment volumes delineated the failure pattern (Figure 1).

Recurrence patterns were defined as follows:Intra-GTV: Recurrence is either entirely contained within the original GTV or originates within the GTV and extends beyond its boundaries.In-CTV/out-GTV: Recurrence occurs within the CTV but does not have any contact with the GTV, indicating that it is completely outside the GTV but still within the CTV.Distant: Recurrence is situated beyond the confines of the radiation field, signifying that the tumor has recurred at a location not encompassed within the original treatment area.Intra-GTV and distant: Recurrence is present both within the GTV and at a distant site outside the radiation field, implying multiple areas of tumor regrowth.

However, for analytical purposes, the failure patterns were categorized into two respective groups (Group 1: the recurrent lesion was intra-GTV or in-CTV/out-GTV; Group 2: the recurrent lesion was distant or intra-GTV and distant).

### 2.4. Systemic Inflammation Response Index Calculation

The SIRI was derived using the original formula [22], [(Neutrophil count × Monocyte count) ÷ Lymphocyte count], based on blood parameters obtained prior to surgery and before the administration of corticosteroids.

### 2.5. Statistical Analysis

The primary objective of this study was to assess the potential correlation between SIRI and failure patterns in patients diagnosed with GB. The secondary objective was to delineate the failure patterns after FLAIR-based RT planning. Continuous variables were presented as the mean ± standard deviation (SD), while categorical variables were expressed as frequencies and percentages. Normal distribution conforming data were subjected to comparison using Student’s *t*-test, whereas the Mann–Whitney U test was employed for non-normally distributed data. The most optimal SIRI cut-off value was determined using a receiver operating characteristic (ROC) curve analysis to see how it relates to the failure pattern groups. For this purpose, the cut-off for SIRI was ascertained at the juncture where the J-index exhibited the highest value. The comparative analysis between the SIRI groups was conducted mainly using the Chi-square test. But when the Chi-square test was deemed unsuitable, Fisher’s exact test was employed instead. Bonferroni correction and related *p*-values were used to minimize false positive results when comparing three or more groups, such as the type of surgery performed.

## 3. Results

### 3.1. Patient Characteristics

We assessed a total of 154 patients who received both TMZ and RT at our institution from March 2009 to May 2023. Eighty-three patients were excluded from the study for the following reasons: a missing follow-up (*n* = 20), short-term hypofractionated RT (*n* = 17), unavailable fitting MRI data in failure (*n* = 14), presenting with gliomatosis cerebri (*n* = 11) or brain stem tumors (*n* = 9), having no recurrence (*n* = 7), and not being able to complete the whole RT course (*n* = 5). Therefore, the study cohort analyzed here consisted of the remaining 71 patients who had recurrence and satisfied all of the inclusion criteria. The median age of the patients was 54 years (range: 20–82 years), and most had an ECOG performance status of 0–1 (88.7%). Nearly two-thirds of the patients had undergone subtotal resection. P53, Ki67, and ATRX mutation status was unknown in about 40% of the patients (Table 1).

### 3.2. Failure Pattern Outcomes

The mean time from RT completion to MRI-documented failure was 13 months (range: 3–47 months). Local failure within the GTV occurred in 40 patients (56.3%), while distant metastases (including both the ‘distant’ and ‘GTV + distant’ categories) were diagnosed in 27 patients (38.1%). The mean maximum distance of recurrence (MDR) was 13.5 mm (range: 3–36.9), with the recurrent lesion extending beyond 15 mm in only 7 out of 49 patients. For 29 (59.2%) out of 49 patients experiencing GTV failure (both ‘in GTV’ and ‘in GTV + Distant’ categories), the recurrent lesion did not extend beyond the GTV as defined by FLAIR imaging (Table 2).

### 3.3. Failure Pattern Association Analysis

Table 3 presents data grouped by failure category. The patient characteristics, radiological findings, and pathological parameters were statistically invariant across all failure groups. Using an ROC curve analysis, we searched for an optimum cut-off that may discriminate between the failure pattern groups. The optimal SIRI cut-off value was determined as 3.03 [area under the curve: 71.8%; sensitivity: 71.8%; specificity: 70.3%; J-index: 0.421) (Figure 2). Our findings reveal that a higher SIRI value indicates higher rates of Group 1 failures (68.6% vs. 44.4%) despite the almost identical distribution of the other variables among the two SIRI groups.

## 4. Discussion

In our cohort of patients diagnosed with IDH wild-type GB and subjected to treatment under our simultaneous integrated boost RT protocol since 2009, we examined failure patterns by calculating the magnitude of the MDR from the target, denoted as GTV. Our treatment protocol encompassed FLAIR-based targeting, administering 60 Gy to the GTV and 50 Gy to a 20 mm encompassing CTV. Our findings reveal that only 9% of failures transcended a 15 mm margin from the FLAIR-based GTV. Importantly, our data suggest lowering the margin to 15 mm around the FLAIR-based GTV, which is consistent with the ESTRO-EANO standards that target no more than 15 mm regions. This reduction strategy appears viable even with a lower dose of 50 Gy, as utilized in our protocol, as opposed to the 60 Gy dose in 30 fractions recommended by the ESTRO-EANO guidelines. Importantly, our research findings reveal a significant correlation between SIRI groups and recurrence patterns in patients with GB treated with concurrent RT and TMZ trailed by adjuvant TMZ. These findings indicate a substantial role for the systemic inflammatory response status as a determinant of recurrence patterns in this patient cohort.

Our study is the first to evaluate the absolute MDR from the target, defined as the GTV, rather than using the traditional definition of failure [23]. While numerous studies have examined failure patterns in patients with GB, they predominantly defined failure based on the isodose curve, classifying failures as in-field, marginal, or distant [23,24,25]. Although Langhans et al. used distance metrics based on the center of recurrence to derive recurrence patterns in their article, they indicated that most recurrences (75%) occurred locally within the primary tumor area [26]. In their study, Tu et al. examined the distance from the center of local cancer recurrences to the edge of the original tumors in patients with recurring GB [27]. They found that a 10 mm margin from T1w-enhanced lesions and a 5 mm margin from T2w/FLAIR abnormal lesions could include 94.8% and 98.3% of local recurrences, respectively. However, it is noteworthy that their study focused on measuring the distance from the central point of the recurrent lesion rather than the MDR used in our study. Measuring the distance from the center of the recurrence to the GTV means excluding the half-diameter of the tumors outside the GTV or extending clearly from the GTV; thus, a lower extension distance is recorded. Hence, we believe that delineating the MDR from the GTV border in our study achieved a more realistic measurement.

Research investigating the reduction in margins to alleviate toxicity in patients with GB, who typically exhibit poor survival rates, has been established for numerous decades. Available investigations have consistently revealed an invariant failure trend when employing limited margins in GB [28,29]. Furthermore, two pivotal studies have illustrated that local failure rates remain comparable even when using more constrained CTV margins [30,31]. In their research, McDonald et al. employed limited-margin RT for GBs and observed that most recurrences were central, comprising 78%, while only 2% were distant [30]. Gebhardt et al. documented the outcomes of 95 patients who were managed in accordance with the Adult Brain Tumor Consortium (ABTC) guidelines involving limited margins. Their analysis revealed that 66% of patients encountered in-field failures exclusively, with 28% experiencing distant failures [31]. The latest guideline emphasizes the importance of using FLAIR-based target delineation to reduce recurrence rates and minimize margins [5]. Moreover, in a recent study, Yilmaz et al. reported that the marginal recurrences that had been delineated based on the EORTC ((ESTRO)-ACROP, 2016 [32]) and RTOG guidelines were within the boundaries of treatment field according to FLAIR-based target identification [33]. Our current study identified recurrences within FLAIR-based GTV in 56.3% of the patient cohort. Notably, 29 (59.2%) out of the 49 patients experiencing intra-GTV failure exhibited no extension beyond the GTV target, while distant recurrences manifested in 38.1% of cases. Despite the challenge of attributing this discrepancy to a specific cause, we interpret the higher incidence of out-of-field recurrences in our current treatment protocol, compared to those reported in the literature, as an indication of the success of local treatment in addressing all FLAIR invasions. Additionally, while further confirmatory studies are required, the use of FLAIR sequences to evaluate newly emerging lesions for the purpose of diagnosing recurrence may elucidate the notable prevalence of distant failures observed in our study.

Our current study’s novel and most significant discovery was the initial demonstration of a valuable utility for SIRI in accurately predicting distinct recurrence patterns in patients with GB. In recent years, there has been a growing interest in the potential use of immune and inflammatory markers as prognostic indicators in patients with GB. Shi et al. conducted an analysis on 232 patients with GB, focusing on preoperative parameters, such as inflammatory markers. These markers included the SII, SIRI, NLR, platelet-to-lymphocyte ratio (PLR), monocyte-to-lymphocyte ratio (MLR), and albumin-to-globulin ratio (AGR) [34]. According to their findings, the SII was the exclusive independent inflammatory marker for accurately predicting patient prognosis. Jarmuzek et al. conducted a study on 358 grade 4 glial tumors to explore the impact of the NLR, SII, and SIRI levels [17]. Their findings revealed a significant association between NLR ≥ 4.56, SII ≥ 2003, and SIRI ≥ 3.03 with an elevated risk of mortality. Wang et al. emphasized the significance of the SIRI in patients with GB through propensity score matching [14]. Furthermore, Topkan et al. identified elevated pre-treatment levels of SII (≥565) and SIRI (>1.78) as distinctive independent prognostic indicators for anticipating survival outcomes [15,16]. A recent meta-analysis involving 21 studies revealed that NLR, PLR, and cell-free DNA (cfDNA) serve as effective peripheral inflammatory markers for prognostic assessment in individuals with GB [17]. Our investigation deviates from previous studies in that it specifically examines the potential of the SIRI to forecast relapse patterns rather than focusing on survival outcomes, which has been the primary emphasis of published research. Notably, a heightened SIRI level surpassing the designated cut-off of ≥3 adeptly anticipated Group 1 failure, characterized by the ‘intra-GTV’ or ‘in-CTV/out-GTV’ pattern. This discernment substantiates that the SIRI not only holds value in predicting survival outcomes but also demonstrates significant usefulness in discriminating between failure patterns among patients with GB. Yet, further investigation is required to validate this innovative finding before it is integrated into standard oncological practice for these patient populations.

Systemic inflammation has been observed to predict prognosis in various cohorts of FLAIR-based diseases. While the existing literature lacks a study specifically evaluating the predictive capacity of the SIRI for determining failure patterns, high-grade gliomas are characterized by distinct features of inflammation. Additionally, the link between gliomas and systemic inflammation has been identified as a significant factor that plays a crucial role in initiating and advancing the gliomagenesis process [35]. The existing literature has examined the association between high-grade gliomas and neutrophils, a crucial component of the SIRI, and it has shown substantially elevated levels of neutrophil infiltration in the most malignant gliomas [36]. Additionally, increased neutrophil infiltration has been identified as one reason for tumor progression and treatment resistance because neutrophils promote the proliferation of glioma stem cells [37,38]. Tumor-associated macrophages and microglia (TAMs) are akin to circulating monocytes but specific to brain tissue. It has been demonstrated in vitro that they enhance the invasiveness of glioma stem cell-like cells through the release of TGF-β1 [39]. TAMs are a plentiful population of immune-inflammatory cells in the GB tumor microenvironment. They play a crucial role in supporting tumor progression, inducing immunosuppression and aggravated inflammation; hence, the infiltration of immunosuppressive TAMs engenders resistance to emerging immunotherapies [40]. Lymphocytes represent the third and final component of the SIRI. In contrast to neutrophils and monocytes, decreased levels of lymphocytes signify compromised immune function, uncontrolled inflammation, and a deteriorating disease prognosis attributable to an exacerbated disease course. [41]. Increased T-cell infiltration is a well-established predictive marker for an enhanced immunogenic effect and improved prognosis, supporting this interpretation [42]. Despite existing uncertainties, the available data all point towards a notable association between a high SIRI level in patients with Group 1 failures and the indication of the tumor’s resistance to treatment. In Group 2, characterized by low systemic inflammation, it may be posited that TMZ was inadequate for distant control. Nevertheless, these remarks should only be valued as rational speculations until they are reinforced by research explicitly addressing these issues.

Despite adhering to consistent diagnostic, planning, and treatment protocols, the present study is subject to certain limitations. First, its small sample size and retrospective design as a single institutional study make it susceptible to selection biases. Second, the cohort does not represent patients treated with hypofractionated regimens or IDH mutant high-grade gliomas. Hence, it is imperative to validate the applicability of our findings in this subset of patients to determine their generalizability across all patients with GB irrespective of potential confounding variables. Third, due to this study’s retrospective nature, we may have missed the opportunity to access more comprehensive pathological details that could have demonstrated correlations with SIRI levels and recurrence patterns. And fourth, our SIRI measurements and the corresponding cut-off values were derived from a single time-point snapshot. However, it is important to note that changes in the counts of the SIRI components can notably impact the optimal cut-off during both the RT plus TMZ and maintenance TMZ phases. Further investigation is warranted to assess the SIRI levels at various time points to ascertain a more precise cut-off value that can more effectively differentiate the failure patterns among patients with GB. Hence, it is critical to regard the current study’s findings as hypothetical rather than definitive recommendations until corroborating research results become available.

## 5. Conclusions

Our research indicates that the SIRI could be a valuable biomarker for predicting specific failure patterns in patients with GB. In addition, our findings support the updated ESTRO-EANO guidelines for defining target volumes in patients with GB, which recommend using 15 mm margins around the GTV to create the CTV. Nevertheless, it is crucial to conduct large-scale prospective studies in the future to validate these results before implementing them in routine radiation oncology practice.

## Figures and Tables

**Figure 1 brainsci-14-00922-f001:**
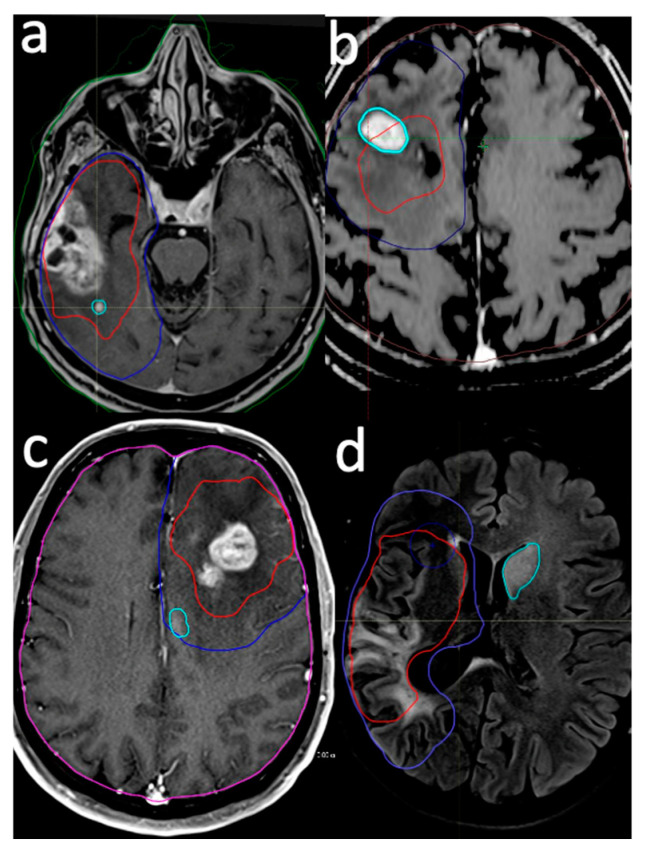
Examples of failure pattern definition. Intra-GTV (**a**,**b**), in-CTV/out-GTV (**c**), and distant (**d**). Red line: gross tumor volume (prescribed dose: 60 Gy/30 fr), blue line: clinical target volume (prescribed dose: 50 Gy/30 fr), cyan line: recurrent lesion.

**Figure 2 brainsci-14-00922-f002:**
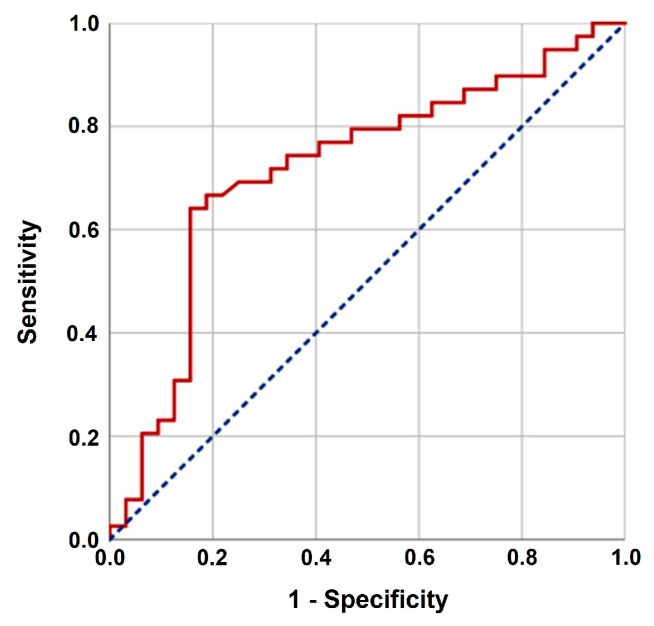
Receiver operating characteristic curve analyses outcomes. Area under curve: 71.8%; sensitivity: 71.8%; specificity: 70.3%; J-index: 0.421.

**Table 1 brainsci-14-00922-t001:** Patient characteristics.

Characteristic	All Patients (N = 71)	SIRI < 3 (N = 36)	SIRI ≥ 3 (N = 35)	*p*-Value
Age group, N (%)				0.49
<50	23 (34.4)	13 (36.1)	10 (28.6)	
≥50	48 (67.6)	23 (63.9)	25 (71.4)	
Gender, N (%)				0.49
Male	48 (32.4)	12 (33.3)	11 (31.4)	
Female	23 (67.6)	24 (66.7)	24 (68.6)	
ECOG, N (%)				0.96
0–1	63 (88.7)	32 (88.9)	31 (88.6)	
2	8 (11.3)	4 (11.1)	4 (11.4)	
Tumor, N (%)				0.56
Unicentric	59 (83.1)	29 (80.6)	30 (85.7)	
Multicentric	12 (16.9)	7 (19.4)	5 (14.3)	
Corpus callosum involvement, N (%)				0.92
Yes	26 (36.6)	13 (36.1)	13 (37.4)	
No	45 (63.4)	23 (63.9)	22	
Thalamic involvement, N (%)				0.32
Yes	20 (28.2)	12 (33.3)	8 (22.9)	
No	51 (71.8)	24 (66.7)	27 (77.1)	
Surgery type, N (%)				0.92
Gross total	15 (21.1)	8 (22.2)	7 (20.0)	
Subtotal	47 (66.2)	23 (63.9)	24 (68.6)	
Biopsy	9 (12.7)	5 (13.9)	4 (11.4)	
P53 mutation status, N (%)				0.89
Yes	20 (28.2)	11 (30.6)	9 (25.7)	
No	20 (28.2)	10 (27.8)	10 (28.6)	
Unknown	31 (43.6)	15 (41.6)	16 (45.7)	
ATRX mutation status, N (%)				0.82
Yes	3 (4.2)	2 (5.5)	1 (2.9)	
No	39 (54.9)	19 (52.8)	20 (57.1)	
Unknown	29 (40.9)	15 (41.7)	14 (40.0)	
Ki 67 status, N (%)				0.56
<30%	18 (25.3)	10 (27.9)	8 (22.9)	
≥30%	25 (35.2)	14 (38.8)	11 (31.4)	
Unknown	28 (39.5)	12 (33.3)	16 (45.7)	

Abbreviations: SIRI: systemic immune response index; ECOG: Eastern Cooperative Oncology Group; ATRX: alpha-thalassemia/intellectual disability, X-linked.

**Table 2 brainsci-14-00922-t002:** Treatment and failure characteristics.

Characteristic	Patients (N = 71)	SIRI < 3 (N = 36)	SIRI ≥ 3 (N = 35)	*p*-Value
Whole brain volume (cm^3^)				0.90
Mean (range)	1356 (861.4–1785.6)	1359 (861.4–1785.6)	1353 (1106–1750)	
GTV (cm^3^)				0.60
Mean (range)	137.6 (14.2–339.3)	138 (14.2–326)	148 (45.4–339.3)	
CTV (cm^3^)				0.46
Mean (range)	333 (85.5–706)	317 (85.5–706)	339 (149–613.53)	
Failure pattern, N (%)				0.20
Intra GTV	40 (56.3%)	16 (44.4)	24 (68.6)	
In CTV/out GTV	4 (5.6%)	2 (5.6)	2 (5.7)	
Distant	18 (25.4%)	12 (33.3)	6 (17.1)	
Intra GTV + Distant	9 (12.7%)	6 (16.7)	3 (8.6)	
Failure group, N (%)				0.035
Group 1	44 (62%)	18 (50.0)	26 (74.3)	
Group 2	27 (27%)	18 (50.0)	9 (25.7)	
MDR, mm				0.75
Mean (range)	13.5 (3–36.9)	1.31 (3–32.5)	13.9 (3–36.9)	

Abbreviations: SIRI: systemic immune response index, GTV: gross tumor volume, CTV: clinical target volume, MDR: maximal distance of recurrence.

**Table 3 brainsci-14-00922-t003:** Failure pattern association analysis.

Characteristic		Failure Pattern Groups		
		Group 1 (Intra-GTV; in-CTV/out-GTV) N = 44 (%)	Group 2 (Distant; Intra-GTV + Distant) N = 27 (%)	*p*-Value
Age, N (%)	<55 years	20 (45.4)	14 (51.9)	0.60
	≥55 years	24 (54.6)	13 (48.1)	
Gender, N (%)	Female	13 (29.5)	10 (37.1)	0.51
	Male	31 (70.5)	17 (62.9)	
Surgery, N (%)	GTR	11 (25)	4 (14.8)	0.13
	STR	30 (68.1)	17 (62.9)	
	Biopsy	3 (6.9)	6 (22.3)	
ECOG, N (%)	0–1	40(90.9)	23 (85.2)	0.75
	2	4 (9.1)	4 (14.8)	
P53 Mutation, N (%)	No	13 (29.5)	7 (25.9)	0.873
	Yes	13 (29.5)	7 (25.9)	
	Unknown	18 (41)	13 (48.2)	
ATRX Mutation, N (%)	No	24 (54.6)	15 (55.6)	0.98
	Yes	2 (4.4)	1 (3.7)	
	Unknown	18 (41)	11 (40.7)	
Ki 67, N (%)	<30%	12 (27.3)	6 (22.2)	0.89
	≥30%	15 (34.1)	10 (37.1)	
	Unknown	17 (38.6)	11 (40.7)	
Tumor, N (%)	Unicentric	37 (84.1)	22 (81.4)	0.77
	Multicentric	7 (15.9)	5 (18.6)	
Corpus callosum involvement, N (%)	No	29 (65.9)	16 (59.3)	0.57
	Yes	15 (34.1)	11 (40.7)	
Thalamic involvement, N (%)	No	33 (75)	18 (66.7)	0.44
	Yes	11 (25)	9 (33.3)	
SIRI, N (%)	<3	18 (40)	18 (66.7)	0.03
	≥3	26 (60)	9 (33.3)	
	Yes	26 (59)	12 (44.4)	

Abbreviations: SIRI: systemic immune response index; ECOG: Eastern Cooperative Oncology Group; ATRX: alpha-thalassemia/intellectual disability, X-linked.

## Data Availability

The presented data belong to and are stored at the Koc University School of Medicine; they cannot be shared without permission. For researchers who meet the following criteria for access to confidential data, please contact the Koc University Committee on Human Research: chr@ku.edu.tr.

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
