# Peer review of "The Systemic Inflammation Response Index Efficiently Discriminates between the Failure Patterns of Patients with Isocitrate Dehydrogenase Wild-Type Glioblastoma Following Radiochemotherapy with FLAIR-Based Gross Tumor Volume Delineation"

_brainsci, 2024, doi:10.3390/brainsci14090922_

Round 1

Reviewer 1 Report

Comments and Suggestions for Authors

Dear authors,

You have done an interesting retrospective study, but there are several points that need clarifications:

1. In lines 66 and 67 you wrote: “The development and progression of gliomas are closely linked to inflammation status and immune response.” This statement is true, but there are other aspects related with GBMs’ resistance to therapy, such as alteration in multiple signalling pathways, intra-tumoral heterogeneity, genetic and epigenetic variations among tumour cells and changes in the tumoral microenvironment. These aspects also need to be discussed.

2. Table 2 show that 38 patients (53.5%) received bevacizumab while in “Patients and Methods” bevacizumab is not described as part of the initial treatment. Bevacizumab is used a part of the systemic therapy of recurrence.

3. In table 3, the number (N) of patients is missing.

4. To further assess the role of SIRI in GBMs’ recurrences, it would be useful to determine its value at the time of recurrence, since radiation therapy, temozolomide y dexamethasone could change SIRI’s value.

5. Pathological studies of the resected GBMs would also be essential to correlate the value of SIRI with the inflammatory state of the tumor.  

Author Response

Reviewers' Comments to the Authors:

Reviewer 1.

Comment 1. In lines 66 and 67 you wrote: “The development and progression of gliomas are closely linked to inflammation status and immune response.” This statement is true, but there are other aspects related with GBMs’ resistance to therapy, such as alteration in multiple signaling pathways, intra-tumoral heterogeneity, genetic and epigenetic variations among tumor cells and changes in the tumoral microenvironment. These aspects also need to be discussed.

Response to Comment 1. Thank you for your suggestion. Exploring this aspect could have been intriguing. We have incorporated the relevant factors you pointed out, complete with their references (see lines: 66-67). Nonetheless, delving into these factors in detail would not be suitable, as our current study is concentrated on inflammation parameters. We believe that expanding the scope to include additional factors would deviate from the central focus of our research.

Comment 2. Table 2 show that 38 patients (53.5%) received bevacizumab while in “Patients and Methods” bevacizumab is not described as part of the initial treatment. Bevacizumab is used a part of the systemic therapy of recurrence.

Response to Comment 2. Thank you for bringing this misunderstanding to our attention, and we apologize for any lack of clarity in our presentation. The usage rate of Bevacizumab, as defined in the table, refers to the percentage of patients who were administered the drug at the time of relapse. This was specified in the section on target delineation, where we mentioned its use during relapse (see lines 116-117). Furthermore, we have updated the caption of this section to “Target Delineation and Systemic Treatment” for greater precision (line 98).

Comment 3. In table 3, the number (N) of patients is missing.

Response to Comment 3. Thank you for pointing this out. The N (%) has been added in Table 3

Comment 4. To further assess the role of SIRI in GBMs’ recurrences, it would be useful to determine its value at the time of recurrence, since radiation therapy, temozolomide y dexamethasone could change SIRI’s value.

Response to Comment 4. We concur with the reviewer that an elaboration on this point with additional data would be beneficial. However, we find that expanding our dataset is not feasible at this time. Accordingly, we have acknowledged this constraint in our study by noting that the single time-point snapshot SIRI calculation is a limitation, as mentioned in line 334.

Comment 5. Pathological studies of the resected GBMs would also be essential to correlate the value of SIRI with the inflammatory state of the tumor.  

Response to Comment 5. We acknowledge the reviewer’s point and have therefore provided a detailed discussion of studies that investigate the status of SIRI parameters in tissue between lines 302-320. However, evaluating these parameters in the pathology specimens from our study’s patients was not feasible. Regrettably, due to the unavailability of pathological details such as P53, Ki67, and ATRX status for most patients, we were unable to establish any correlation between the SIRI cut-off value and patterns of recurrence. We have included this limitation in the relevant section of our manuscript (see lines 331-333).

Reviewer 2 Report

Comments and Suggestions for Authors

The authors of the original paper entitled "Systemic Inflammation Response Index Efficiently Discriminates Between the Failure Patterns of IDH-Wild Type Glioblastoma Patients Following Radiochemotherapy of Flair-Based GTV Target Volume Delineation" address a complex and, at the same time, innovative topic for the treatment of a terrible tumor, such as Glioblastoma Multiforme. The study focuses on the identification of the systemic inflammatory response index (SIRI) to stratify patients with patterns of failure in IDH-wild type GBM.

Major revision:

·      It would be interesting to insert a table that examined the associtaion between high-grade gliomas and neutrophils and high-grade gliomas and lymphocytes ratios for patients employed in this study, in order to obtain more information regarding the key components of SIRI.

·      It is recommended to include data regarding the ethics committee approval for the study performed

Minor revision:

- line 50 and line 54: specify what is meant by "T2" and "T1"

- it is recommended to insert a comma before "such as"

- an update of the references is recommended, as there are no references of 2024 for example

Author Response

Response to Reviewer 2

Major revision:

Comment 1:  It would be interesting to insert a table that examined the association between high-grade gliomas and neutrophils and high-grade gliomas and lymphocytes ratios for patients employed in this study, in order to obtain more information regarding the key components of SIRI.

Response to Comment 1: While we appreciate the reviewer’s feedback, we respectfully elaborate that point why we did not prefer to insert a table for neutrophils and lymphocytes ratios for patients. We did not prefer to make a separate table for neutrophils and lymphocytes ratios (NLR) because SIRI already includes NLR and is a more comprehensive inflammation marker than NLR. In addition, creating a separate table for this parameter requires editing the discussion, results and method sections and it is necessary to reestablish the hypothesis of the study. However, we preferred not to make this hypothesis change for a marker that includes fewer parameters than SIRI.

Comment 2: It is recommended to include data regarding the ethics committee approval for the study performed

Response 2: Thank you for pointing this out. The ethics committee approval number had been added in ‘’Institutional Review Board Statement section’’ (Line 364). The document will send separately.

Minor revision:

Comment 1 line 50 and line 54: specify what is meant by "T2" and "T1"

Response 1. We’ve made the change as T2-weighted images (T2w) and T1-weighted postcontrast (T1w) in related lines.

Comment 2 it is recommended to insert a comma before "such as"

Response 2: Thank you for your attention. The grammatical mistakes were revised in lines 73 and 275. 

Comment 3: an update of the references is recommended, as there are no references of 2024 for example

Response 3. Thank you for this suggestion that is important to indicate the up-to-dateness of the study. References are updated and recently published four references were added (Line 68 and 309).

Round 2

Reviewer 1 Report

Comments and Suggestions for Authors

Dear authors,

You have improved the introduction and the discussion of the manuscript, but there is an important methodological aspect that needs further explanations. As the authors explain in the introduction: “… our current study aimed to assess the patterns of failure in IDH-wild type GBs as per the aforementioned protocol, and to examine the discriminative capacity of SIRI across diverse brain relapse patterns.” And also, according to the authors: “Bevacizumab was not used as initial treatment. Its use was allowed in relapsed disease”. Then the 38 patients treated with bevacizumab, more than half the patients of the study, were relapsed patients, that were not receiving the first line treatment.

Following this line of thought, to evaluate the role of SIRI as a prognostic factor for the response rate to the treatment with radiation therapy and temozolomide, all bevacizumab treated patient, which mean relapsed patients, cannot be part of this statistical study. 

Author Response

Comment: You have improved the introduction and the discussion of the manuscript, but there is an important methodological aspect that needs further explanations. As the authors explain in the introduction: “… our current study aimed to assess the patterns of failure in IDH-wild type GBs as per the aforementioned protocol, and to examine the discriminative capacity of SIRI across diverse brain relapse patterns.” And also, according to the authors: “Bevacizumab was not used as initial treatment. Its use was allowed in relapsed disease”. Then the 38 patients treated with bevacizumab, more than half the patients of the study, were relapsed patients, that were not receiving the first line treatment. 

Following this line of thought, to evaluate the role of SIRI as a prognostic factor for the response rate to the treatment with radiation therapy and temozolomide, all bevacizumab treated patient, which mean relapsed patients, cannot be part of this statistical study. 

Response: We would like to thank you for pointing out the issue regarding Bevacizumab usage, which helped us realize the missing information. In fact, all patients included in the study were relapsed patients, as emphasized in the methods section. Some received Bevacizumab, while others received chemotherapy or radiotherapy during their relapse. The failure results presented in the study reflect the outcomes of initial treatments prior to the administration of Bevacizumab. However, we initially presented only the Bevacizumab usage data, which, due to missing information, led to some confusion. Therefore, we decided to remove the data related to radionecrosis and Bevacizumab usage from the table, aiming to improve clarity without altering the design or results of the study. The referee's comments also indicated that including this information caused the study to deviate from its primary focus.

We have also revised the introduction and discussion based on your suggestion. We have added a paragraph in the introduction stating the importance of FLAIR (Lines 67-72). Also, we have developed discussion section and insert new literature about failure pattern (Lines 245-248 and 268-273).